# Evolutionary Specializations in the Venous Anatomy of the Two-Toed Sloth (*Choloepus didactylus*): Insights from CT-scan 3D Reconstructions

**DOI:** 10.3390/ani14121768

**Published:** 2024-06-12

**Authors:** Paul Martre, Baptiste Mulot, Edouard Roussel, Antoine Leclerc

**Affiliations:** 1Hôpital Privé de l’Estuaire, 76620 Le Havre, France; 2ZooParc de Beauval and Beauval Nature, 41110 Saint-Aignan, France; baptiste.mulot@beauvalnature.org (B.M.); antoine.leclerc@zoobeauval.com (A.L.); 3Department of Digestive Surgery, Rouen University Hospital, 76000 Rouen, France; edouard.roussel@chu-rouen.fr

**Keywords:** *Choloepus didactylus*, sloth, xenathrans, caudal vena cava, 3D reconstruction, 3D Slicer, CT scan, anatomy, veins

## Abstract

**Simple Summary:**

There are few available data on the venous anatomy of Choloepus didactylus, particularly in living subjects. This study highlights the unique venous anatomy of this species in a non-invasive, retrospective, and reproducible manner, to enhance our understanding of this understudied species.

**Abstract:**

The venous anatomy of the two-toed sloth (*Choloepus didactylus*) remains poorly understood, particularly in living specimens due to the limitations of traditional cadaveric studies. This study aims to describe the unique venous structures of *Choloepus didactylus* using computed tomography, enhancing our understanding of this species in a live setting. Three living *Choloepus didactylus* underwent CT scans as part of routine clinical assessments. The images were reconstructed using 3D Slicer software (version 5.6.2), focusing on the caudal vena cava anatomy. The reconstructions confirmed the presence of a significant intravertebral vein, showing complex venous connections through the ventral sacral foramen and vertebral foramina. These findings highlight notable anatomical variations and challenge existing literature on the species’ venous architecture. By employing modern imaging technologies, this research provides new insights into the venous anatomy of *Choloepus didactylus*, demonstrating the value of non-invasive techniques in studying the anatomical features of live animals, thereby offering a foundation for further comparative and evolutionary studies.

## 1. Introduction

Described by Buffon as “the final term of existence in the order of animals”, which “one more defect would have prevented from subsisting” [1], sloths are a unique species of mammal in many ways. These xenarthran mammals constitute the suborder Folivora (order Pilosa) and include two-toed species of the genus *Choloepus* and three-toed species of the genus *Bradypus*, which diverge significantly (Figure 1: images from phylopic.org).

Living at its own pace in the canopy of tropical rain forests, *Choloepus didactylus*, Linné’s two-toed sloth, is a distinct illustration of evolutionary specialization. Two-toed sloths have adapted to a nocturnal lifestyle, exhibiting unique physiological and behavioral traits setting them apart from their diurnal counterparts and other mammals [2,3]. Yet, available data on *Choloepus didactylus* anatomy and physiology remain limited.

The caudal vena cava of sloths was described by Wislocki in 1928, based on a cadaveric study involving about 50 sloths (three-quarters being three-toed sloths, *Bradypus griseus*, now named *Bradypus variegatus*, and one-quarter being Hoffmann’s two-toed sloths, *Choloepus hoffmanni*) [4].

This study described, in addition to the known double caudal vena cava of xenarthrans, an unusual reduction in the diameter of the caudal vena cava in its cranial abdominal part: “In spite of the fact that in its course the vena cava has received renal, spermatic (i.e., testicular), and adrenal tributaries, it dwindles to almost nothing in its upper course before it reaches the liver. Only within the liver does it again become augmented in caliber in its upper part”.

In *Choloepus*, there was also a description of circulation within the vertebral canal via a “tremendous venous trunk” that deflects a large part of the blood flow from the pelvic limbs and kidneys. This venous trunk of great importance connects to the double caudal vena cava through relatively large vessels entering the intervertebral foramina and then anastomoses with the cranial vena cava via the proximal portion of the azygos vein before reaching the right atrium [4]. The azygos vein, except for its proximal end, and the hemiazygos vein are absent. Conversely, no significant reduction in the diameter of the pre-hepatic segment of the caudal vena cava was reported in Bradypus variegatus, and the diameter of the vertebral venous trunk appeared reduced in this species.

Anastomoses between the caudal vena cava and the intravertebral veins via two larger veins (*venae basivertebrales*) [5] which perforate the bodies of the lumbar and thoracic vertebrae via vertebral foramina were described by De Burlet at the beginning of the 20th century [6].

These observations were later confirmed by the research of Barnett et al. in 1957 [7] and then by Hoffstetter in 1959 [8] in both *Choloepus* and *Bradypus*: “Thanks to this remarkable mechanism, a significant portion of the blood from the posterior part of the body returns to the heart not through the inferior (i.e., caudal) vena cava, but through the superior (i.e., cranial) one”.

Nevertheless, two more recent works simply describe a double caudal vena cava system, without mentioning the intravertebral venous system [3,9].

Another cadaveric study published in 2021, involving three specimens of *Choloepus hoffmani*, even refutes this assertion, stating that the “irrigation of the heart is consistent with that of domestic mammals” [10].

Such contradictory findings resulted in a controversy regarding the caudal cava venous system of sloths, particularly the two-toed sloths. The only available data on the two-toed sloths concern *Choloepus hoffmanni* and are based solely on cadaveric series.

Recent studies have demonstrated the value of CT-scan studies for investigating anatomical variations, both for illustrating variations within the same species [11,12] and differences between species [13]. The two main techniques available are volume rendering and 3D reconstruction. Volume rendering is a technique that allows for the visualization of images in three dimensions by mapping the opacity and color of different Hounsfield unit values to optimize the visualization of desired structures. The advantage of this technique is that it allows for real-time, interactive visualization of structures. However, it does not allow for fine measurements or image post-processing. On the other hand, 3D reconstruction enables the creation of 3D models that can be used for more precise measurements, and these models can also be exported to post-processing software [14,15]. The reconstruction process can be complex and time-consuming, but it provides images that are not always possible to obtain with the volume rendering technique.

The purpose of the study was to describe the anatomy of the caudal vena cava (VC) in *Choloepus didactylus*, using 3D reconstruction from three distinct living subjects without performing any invasive procedures.

## 2. Materials and Methods

### 2.1. Animals

Three specimens of *Choloepus didactylus* underwent a CT-scan examination as part of clinical investigations. Subject 1 was a 17-month-old, 6.1 kg female which presented an abscess of the left pelvic limb. Subject 2 was a 10-year-old, 7.5 kg female with respiratory disease that underwent a CT scan to monitor the healing of pulmonary lesions following medical treatment. Subject 3 was a 10-year-old, 5.8 kg male which was examined for severe apathy due to bronchopneumonia.

### 2.2. Computed Tomography Protocol

All acquisitions were performed using a Philips Brilliance 64 CT scan (Philips Medical Systems Nederland B.V., Veenpluis 4–6, 5684 PC Best, the Netherlands). The acquisition parameters for acquiring native DICOM images are summarized in Table 1.

Protocols requiring an IV injection of contrast medium were performed using a Bayer Medrad Salient injection system (Bayer HealthCare SAS, 10 place de Belgique, CS40024, La Garenne Colombes Cedex, France) and iodohexol contrast medium (Omnipaque 350, 350 mg/mL, GE HealthCare SAS, 283 rue de la manière, 78530 Buc, France) with a volume of 2 mL/kg injected at a constant rate of 3 mL per second.

### 2.3. Three-Dimensional Reconstructions

DICOM images of the sloths were imported into 3D Slicer software (version 5.6.2) [16], and segmentation was performed, using the “segment editor” module.

The following tools were used:-Threshold: to enable the selection and assignment of all voxels of a given intensity to the selected segment, or to establish an editable intensity range that can be used in combination with other tools.-Paint and Erase: to assign or remove the desired voxels in the selected segment.-Scissors: to cut through the entire segment from the current viewpoint. Allows one to add or erase voxels from the selected segment.-Smoothing: to enhance the visibility of the selected segment by removing extrusions and filling small holes, making segment boundaries smoother.-Grow from seeds: to grow segments and create complete segmentation, according to location, size, and shape of initial segments. Final segment boundaries were placed where source volume Hounsfield units changed abruptly.

Bones were reconstructed using the threshold tool, manual editing (paint, erase, scissors tools), and smoothing. Liver and kidneys were segmented using the threshold tool, manual editing, and the smoothing tool. Vessels and cardiac cavities were segmented using threshold, grow from seeds, and manual editing.

## 3. Results

Three contrast-enhanced CT scans were available for interpretation. The results are presented in Table 2.

The different portions of the caudal vena cava will be described as follows, from the most cranial portion to the most caudal portion [5,17]:
Post-hepatic segment;Hepatic segment;Pre-hepatic segment, formed by the cranial segment to the renal veins, the renal segment, and the caudal segment to the renal veins.

The large venous communication between the caudal double vena cava and the arch of the azygos will be named the “intravertebral vein”, as described in the works of Wislocki [4], de Burlet [6], and Barnett [7].

The veins passing through the vertebral foramina to join the intravertebral vein will be called the basivertebral veins (*venae basivertebrales*) [5].

Figure 2 shows an overview of each subject. Light blue arrows show the termination of the caudal vena cava in subjects 1 and 2; red arrows show the persistence of the caudal vena cava in subject 3.

Figure 3 shows a right lateral view of each subject. Light blue arrows show the termination of the caudal vena cava in subjects 1 and 2; red arrows show the persistence of the caudal vena cava in subject 3. Green arrows show the anastomoses between the intravertebral vein and the cranial vena cava before reaching the right atrium.

Figure 4 shows a pelvic view of each subject, highlighting the double caudal vena cava and the communications between the intravertebral vein through the basivertebral veins and the ventral sacral foramen.

Figure 5 shows a ventral view of subject 1, highlighting the vertebral foramina. The blue arrow indicates the vertebral foramina located on the third lumbar vertebra.

### 3.1. Subject 1

In the first subject, we observed the presence of a venous network originating from the kidneys, at the level of the 3rd lumbar vertebra, converging at the 23rd thoracic vertebra, and continuing its path cranially up to the 18th thoracic vertebra, to the right side of the spine, before terminating (Figure 2 and Figure 3).

This double caudal vena cava communicated with the intravertebral vein through several basivertebral veins, entering through the vertebral foramina (Figure 4 and Figure 5).

A significant part of the venous flow entered the intravertebral foramen through the second ventral sacral foramen, after receiving the veins from the pelvic limbs (Figure 2, Figure 3, Figure 4 and Figure 5).

This subject did not exhibit any visible communication between the pre-hepatic segment of the caudal double vena cava and the post-hepatic segment of the caudal vena cava draining the supra-hepatic veins (Figure 2 and Figure 3).

After entering the vertebral canal through the first and second ventral sacral foramen and vertebral foramina, the venous flow traveled in the right part of the vertebral canal, to the right of the spinal cord, and then described an anastomotic arch around the ninth right rib, and directly drained into the cranial vena cava, via the arch of the azygos vein (Figure 3)

The right atrium received the post-hepatic segment of the caudal vena cava caudally. Cranially, it received the confluence between the cranial vena cava and the arch of the azygos vein, which itself received the intravertebral vein (Figure 3).

Figure 6 offers a 3D representation of the communications between this intravertebral vein and the double caudal vena cava, through the first and second ventral sacral foramen and the vertebral foramina. This figure was made from the CT-scan reconstruction of subject 1, modified with Blender^®^ software (version 4.0) to enhance quality [18]. Appendix A shows an animation of subject 1, highlighting the position of the caudal vena cava, in the vertebral canal, on the right side of the spinal cord.

Figure 7 shows a CT-scan reconstruction of subject 1, which also underwent a non-contrast CT scan in a suspended position. This image shows the position of the lungs (A), upper digestive system (B), bladder (C), kidneys (D), and rectum (E), in actual anatomical conditions.

### 3.2. Subject 2

The second subject presented a similar anatomy, with a double caudal vena cava originating at the level of L3, merging at the level of T23, and terminating at the level of T18 (Figure 2 and Figure 3).

It also presented basivertebral veins that drained into the vertebral foramina, as well as a communication between the renal veins and the veins of the pelvic limbs, draining into the intravertebral vein via the second and third ventral sacral foramen (Figure 4).

There was also no communication between the pre-hepatic segment of the caudal double vena cava and the post-hepatic segment of the caudal vena cava (Figure 2 and Figure 3).

### 3.3. Subject 3

This subject presented a generally similar disposition of the caudal vena cava.

However, a clear communication between the pre-hepatic segment of the caudal vena cava and the post-hepatic segment of the caudal vena cava was observed, through a small-caliber vein running along the right side of the spine (Figure 2 and Figure 3) [7].

Communications between the intravertebral vein and the arch of the azygos vein occurred through four venous anastomoses in this subject (Figure 3).

## 4. Discussion

All three *Choloepus didactylus* specimens included in this report had a large-caliber intravertebral vein draining the pelvic limbs and kidneys. Our findings corroborate those found in the studies from the 20th century [4,8] and refute the conclusions of more recent works [10].

Occasionally, there may be a communication between the post-hepatic segment of the caudal vena cava and the pre-hepatic segment of the caudal vena cava, as described in 1957 by Barnett’s team, who managed to trace the cranial part of the abdominal caudal vena cava up to the liver in a specimen of *Choloepus* [7].

To the authors’ knowledge, sloths are the only mammals possessing this characteristic.

According to Wislocki, this peculiarity could not be solely explained by the animal’s posture, especially since fossils of terrestrial ancestors of sloths (*Scelidotherium*, *Mylodon*, *Grypotherium*) show traces of intravertebral circulation [4], including the presence of ventral foramina in the vertebral bodies in *Diabolotherium nordenskioldi* [19]. Notwithstanding, these are fossil species, and knowledge of their way of life is, by definition, limited.

The Xenarthra are known to bear several unique anatomic traits, notably paired postrenal venae cavae, as well as additional (xenarthrous) joints of the lumbar vertebrae, and the presence of rete mirabile in the limbs [3,20]. This duplication of the caudal vena cava, also observed in monotremes and many aquatic mammals, results from the persistence of both supracardinal veins in their caudal part, with the caudal vena cava unifying only at the level of the kidneys. This configuration constitutes an anomaly in other species [5,17]. 

In the *Choloepus didactylus* subjects we studied, in addition to this double caudal vena cava, there is, as we have shown in the results, an absence or atrophy of the hepatic part of the caudal vena cava [17].

*Choloepus didactylus* presents several anatomical, physiological and behavioral characteristics that could explain its atypical circulation. It possesses between 23 and 24 thoracic vertebrae and as many ribs, and only 3 lumbar vertebrae [21], which could explain a decrease in abdominal compliance, due to an increase in the rigidity of the dorsal and lateral abdominal walls. 

It has a slow metabolism [22], combined with a unique digestive physiology and anatomy, as its foregut can account for 20 to 30% of its body mass, and it can eliminate up to 250 g of feces and 1200 mL of urine at a time [23]. 

Sloths are known to hang upside down in trees, and *Choloepus didactylus* spend up to 84.3% of their time in this position, beneath branches [2]. 

Due to these characteristics, a caudal vena cava in its usual anatomical position would likely be compressed by the intra-abdominal organs against the spine (Figure 7), and such a compression could probably be intensified by decreased abdominal compliance due to the numerous ribs.

Internal vertebral venous plexuses are well described and known for their compensatory role in humans in cases of chronic obstruction of the caudal vena cava circulation, via horizontal diversion pathways (foraminal anastomoses of the internal vertebral plexuses) and vertical ones (axial contingents of the vertebral venous plexuses) [24]. 

This venous network, which in other species plays a role in diverting venous flow only in pathological cases, may have taken on a predominant role in the renal and pelvic limb venous return in *Choloepus didactylus*, due to a unique metabolism and a resting position different from that of most species. 

Findings from the present report should be compared with other xenarthrans, in particular *Choloepus hoffmanni*, and *Bradypus* spp. [2,25,26], since data from the literature suggest inter- and intra-specific anatomical variability. 

Whether the anatomical characteristics depicted in this report were congenital or acquired remains unclear. Notably, in their juvenile phase, sloths predominantly adopt a ventral decubitus posture, resting on the abdomen of their maternal figure [3]. It would be relevant to acquire tomographic data from juvenile specimens to determine if there is a progressive atrophy of the hepatic segment of the caudal vena cava over the lifetime. 

Limitations of this study include the low spatial resolution of the DICOM images we used during the 3D reconstruction phase, due to the small size of the animals, and the timing of the injection, which sometimes made it difficult to perfectly visualize small-caliber vessels, particularly in subject 3. Furthermore, the limited number of subjects available for analysis in this research is not sufficient to firmly establish morphological statistics. 

However, to the authors’ knowledge, this anatomical study is the first to be conducted on living sloths, thereby avoiding the use of invasive techniques to obtain anatomical data and maintaining physiological conditions during contrast agent injection and acquisition.

## 5. Conclusions

This research highlights unique features in the venous system structure of *Choloepus didactylus* sloths. The use of computer-assisted tomography and 3D reconstruction allowed us to describe the peculiar anatomical configurations of the *Choloepus didactylus*’ caudal vena cava, which differ notably from previous descriptions and from other mammals. These findings not only improve our understanding of the vascular anatomy of sloths but also underscore the importance of modern technology in the anatomical study of poorly documented species.

Moreover, the study raises questions regarding the evolution of these vascular structures in sloths and their possible adaptation to an arboreal and inverted lifestyle.

## Figures and Tables

**Figure 1 animals-14-01768-f001:**
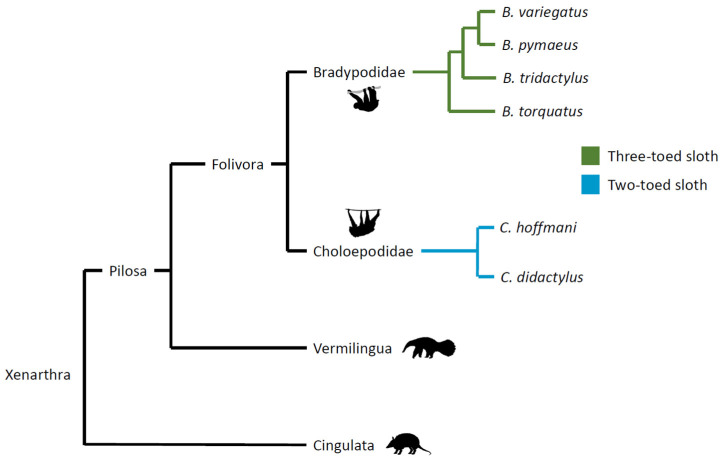
Phylogeny of Xenarthra.

**Figure 2 animals-14-01768-f002:**
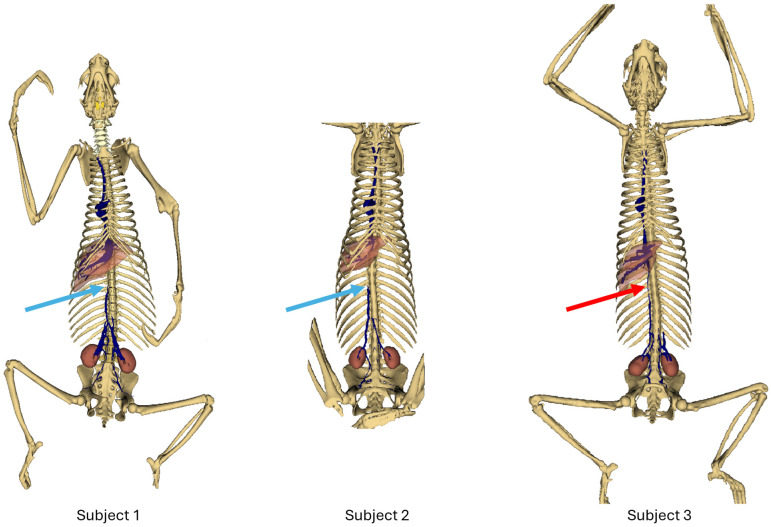
Bones and venous system 3D reconstruction.

**Figure 3 animals-14-01768-f003:**
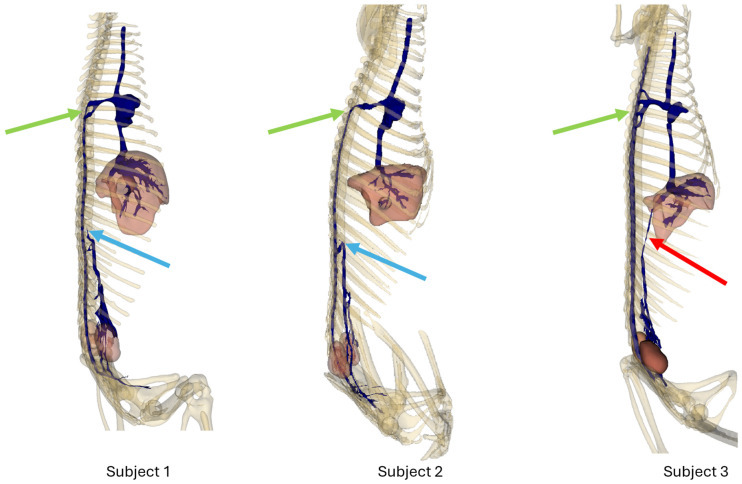
Bones, venous system, kidneys, and liver 3D reconstruction, right lateral view.

**Figure 4 animals-14-01768-f004:**
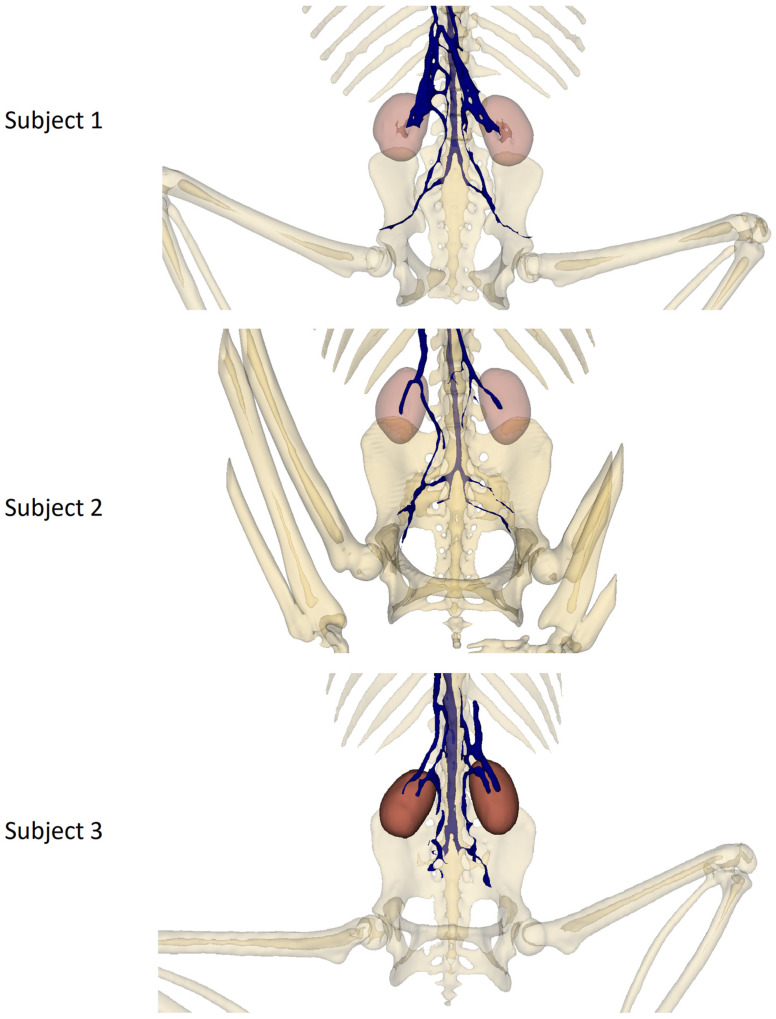
Pelvic bones, kidney, and venous system 3D reconstruction.

**Figure 5 animals-14-01768-f005:**
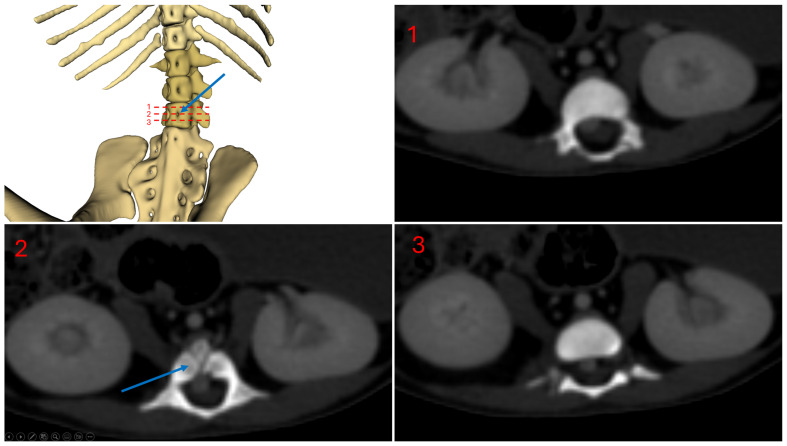
Subject 1 vertebral foramina, L3 level.

**Figure 6 animals-14-01768-f006:**
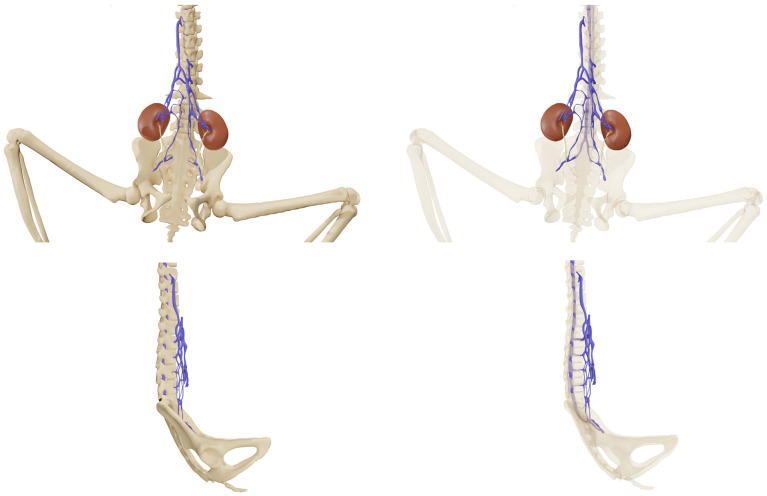
Three-dimensional representation of venous circulation.

**Figure 7 animals-14-01768-f007:**
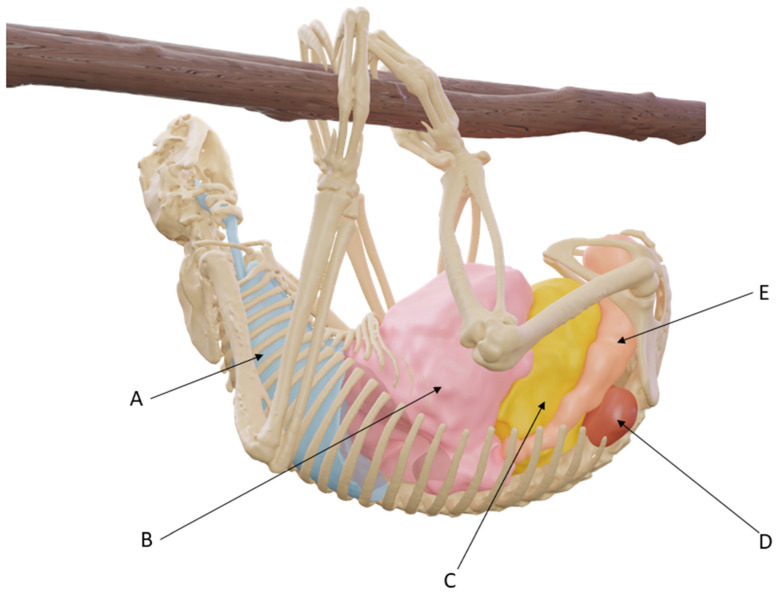
Three-dimensional reconstruction of subject 1 after CT scan in suspended position.

**Table 1 animals-14-01768-t001:** Acquisition parameters.

Subject	Acquisition	mAs	kV	Pitch	FOV (mm)	Rotation Duration (s)	Thickness (mm)	Increment (mm)
Subject 1	Bone	160	120	0.671	150	0.75	0.67	0.4
	Thorax	131	120	0.797	350	0.75	2	1.5
	Thorax IV	132	120	0.797	350	0.75	2	1.5
Subject 2	Thorax	134	120	0.797	350	0.75	2	1.5
	Thorax IV	130	120	0.797	350	0.75	2	1.5
Subject 3	Thorax	166	120	0.797	350	0.75	2	1.5
	Abdomen	206	120	0.797	350	0.75	2	1.5
	Abdomen IV	211	120	0.797	350	0.75	2	1.5

**Table 2 animals-14-01768-t002:** Caudal vena cava characteristics.

Subject	Level of Double Caudal Vena Cava Origin	Presence of Large Caliber Intravertebral Vein	Presence of Communication between the Intravertebral Vein and the Cranial Vena Cava through Ventral Sacral Foramen and Vertebral Foramina	Level of Termination of the Double Caudal Vena Cava	Visible Communication between Double Caudal Vena Cava and Hepatic Veins	Number of Visible Anastomoses between the Intravertebral Vein and the Cranial Vena Cava before Reaching Right Atrium
Subject 1	L3	Yes	Yes	T18	No	2
Subject 2	L3	Yes	Yes	T18	No	1
Subject 3	L3	Yes	Yes	N/A	Yes	4

## Data Availability

All data used in this article are available on figshare (https://figshare.com/s/f7af40638cd24f907908, accessed on 20 May 2024) or by email upon request. Please send an email to the authors to request access to the data.

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
