# Peer review of "Evolutionary Specializations in the Venous Anatomy of the Two-Toed Sloth (Choloepus didactylus): Insights from CT-scan 3D Reconstructions"

_animals, 2024, doi:10.3390/ani14121768_

Round 1

Reviewer 1 Report

Comments and Suggestions for Authors

The paper submitted by the authors provides relevant information about the course of this important vessel in the Two Toed-Sloth. However and before acceptance, you should pay special attention to the different figures included n the text. Then, you should labeled and add arrows pointing pivotal findings. Additionally, it is important to position the figures in a correct anatomical position.  
Specific comments:

Please change “anterior” vena cava by “cranial…“. Please revise the references, then  [2-4] instead of [2], [4]. Lines 50-51, you should revise this paragraph quite carefully since this findings (the reduction) is quite common in other mammals, including dog and cats Line 54, please since it is a quite important finding you should add the reference. Lines 73-78, since modern imaging techniques have demonstrated of  great importance to study anatomical variations, you should explain better this issue. In addition, it is important to explain what kind of  3D reconstruction techniques were used. In Table 1, you include “acquisition parameters”, what do you mind? I suppose that you are referring to the different algorithms used. Lines 129-139, Figure 2? Figure 3, this figure is not well positioned, please put it on a correct anatomical position. Line 146, frontal view is not the correct terminology. Again, it is not a human. Lines 237-239, it is a quite important statement, then add references. Line 237, this clarification is pivotal since duplication of caudal vena cava occurs in pets. Then, you should explain it better. Line 242, I do not agree with this statement due to excellent resolution is acquired in much smaller animals. Additionally, you can use MPR images in order to better visualize these structures.

Reviewer 2 Report

Comments and Suggestions for Authors

I had read the manuscript with interest. While the aim of this study is important, there are significant mistakes and proofs of the lack of anatomical skills and knowledge. The morphological variations and their relations with adaptations of circulatory system for pathological changes are very complex and can guide to vague or ambiguous conclusions, especially when the level of anatomical elaboration of seen features is really low. Dealing with circulatory system morphology and its comparative anatomy, especially venous system, is really hard. Without the deep anatomical knowledge and experiences in such studies (due to the not fully stabilized anatomy of venous blood vessels, which can show numerous variations even in one individual in case of paired vessels), the correct interpretation of morphological features is simply impossible. Moreover, the lack of correct anatomical terminology, including some crucial inconsequence in description, false directional terms typical for basic topographical anatomy and the lack of classical dissections do not allow for accepting the manuscript in current form. I suggest to start from the beginning again. Obtaining the cadavers for anatomical dissection, confronting the achieved findings with other species morphology and classical pattern of venous blood vessels (and nomenclature) in domestic mammals. Secondary, the CT study can be compared with earlier findings to confront the differences, to interpret the visible morphological variations and to form final conclusion. The discussion has a paragraph, which I indicated as a starting point of potential explanation of identified species differences. In my opinion, the work must be remake and improved using the anatomical terminology and methods. Radiology without anatomy does not exist. There is no anterior and posterior vena cava, only in humans we can indicate the superior and inferior one. What is the supra or sub hepatic? Why authors try to give new names of observed structures instead of comparing with existing blood vessels of known names? I hope these remarks will allow for better elaboration of this interesting subject in future. Maybe, the collaboration with well experienced anatomist would be useful?

In this form I suggest to reject the article.

Reviewer 3 Report

Comments and Suggestions for Authors

The presented paper "Evolutionary Specializations in the Venous Anatomy of the Two-Toed Sloth ..." presents the results of 3 individuals that went through a CT scan on their abdominal area. The attempt- that, in my opinion, should not be listed as an "Article" but as a "Case study" is quite an interesting piece of investigation on a species that is little studied in many respects.

Although me, as a Veterinary Anatomist I find the core of this study more than interesting, I found some majour issues...here they are listed as follows and thus the reason on my decision (that should not be taken personally) :

1. in this present form I think the paper should not be published as it lacks a clear perspective from a veterinary anatomist. The study is, most probably, an interpretation of some specialized fellows from human imagining Departments (as I saw the authors in the list) that have no solid knowledge of a series of Veterinary Anatomy details. This is the reason that I stringly advice them to approach a Veterinar Anatomist to clear many aspects of their output.

The nominations used are not appropriate- they should use NAV (Nomina Anatomia Veterinaria) instead of regular AT (Anatomical terminology)(that in fact is not used- they use common names instead of Anatomical/latin names for such designations)- eg. inferior vena cava/ caudal vena cava etc- this is a key element in any anatomical study!

On the other hand, cited sources do not mention just some papers on the sloth anatomical features, but it has no global perspective at all! This should refer as many of the xenarthra group instead of comparing to known human anatomy data (in which the double vena cava caudalis is of course a significant malformative change)

If authors would have gone through some major veterinary anatomical sources  they would have seen that this kind of morphological variations (double pelvic crural veins/ double vena cava, azygous veins and most importantly the frequently invoked communications with the vertebral venous sinuses- from subsacral to lumbar and thoracic intercostal veins are listed not as "unusual" in some species. Please keep in mind that up to the lumbar tributaries, there is large amount of variability in animal kingdom (as mentioned, it has to dio with the development of those cardinal veins). This is why my major advice is to check on the French author (!) that wrote what is known as  "bible of veterinary anatomy"- Robert Barone , and please look for the cahpter of Angiology- the part that deals with radicles and affluents of vene cava caudalis and the azygous vein..you will find there sufficient explanations for very similar situations- (even cetaceans, but also dog, cat, swine)- Barone (1969!), tome 5 Angiologie pages 600-650 (+/-)

So these being said, I am stressing again that the study itself is an nice and good one, but it lacks the key elements that I listed earlier- usage of NAV and veterinary anatomy referencing.

This is why, in my opinion, editors should encourage authors to adapt their study (case study) to these demands/requests..and in this way a very nice report may come to light..

Author Response

Please see attachement. 

Round 2

Reviewer 2 Report

Comments and Suggestions for Authors

The manuscript was significantly improved. All reviewer remarks were used in text. I suggest to accept the manuscript after minor revison (latin nomenclature gramatic remark).

Reviewer 3 Report

Comments and Suggestions for Authors

I am satisfied with the changes that were made by the authors of the study. 

I appreciate the study itself as it brings the new data in regard to the studied species and the additions make it more valuable
